# Occupational, Transport, Leisure-Time, and Overall Sedentary Behaviors and Their Associations with the Risk of Cardiovascular Disease among High-Tech Company Employees

**DOI:** 10.3390/ijerph17103353

**Published:** 2020-05-12

**Authors:** Mei-Lan Liu, Chia-Hui Chang, Ming-Chun Hsueh, Yi-Jin Hu, Yung Liao

**Affiliations:** 1Department of Health Promotion and Health Education, National Taiwan Normal University, 162, Heping East Road Section 1, Taipei 106, Taiwan; 80205005e@ntnu.edu.tw (M.-L.L.); 60605005e@ntnu.edu.tw (C.-H.C.); 2Graduate Institute of Sport Pedagogy, University of Taipei, Taipei 11153, Taiwan; boxeo@utaipei.edu.tw; 3Adjunct researcher, Faculty of Sport Sciences, Waseda University, 2-579-15, Mikashima, Tokorozawa City Saitama Prefecture 359-1192, Japan

**Keywords:** cardiovascular disease, sedentary behavior, high-tech company, occupational health

## Abstract

This study examined the associations of overall and domain-specific (i.e., occupational, transport, and leisure-time) sedentary behaviors with cardiovascular disease (CVD) risk factors among high-tech company employees in Taiwan. A total of 363 participants employed at high-tech companies (mean age ± standard deviation: 37.4 ± 7.2 years) completed a questionnaire administered by email regarding their overall, occupational, transport, and leisure-time sedentary behaviors. Self-reported data of height and weight, blood pressure, blood sugar, and total cholesterol levels were also collected in 2018. An adjusted binary logistic regression model was employed in the analysis. After adjusting for sociodemographic variables, high-tech company employees who used a computer (or Internet) for more than 2 h per day during their leisure time were more likely to have CVD risk factors (odds ratio: 1.80; 95% confidence interval: 1.08–3.00). No significant associations with CVD risk factors were detected for total sedentary time, occupational sitting, television viewing time, and transport-related sitting. Despite the nature of cross-sectional design in this study, our findings may have considerable implications for intervention designers and policymakers of Taiwan. Developing effective strategies for limiting leisure-time computer use should be considered for the prevention of CVD among high-tech company employees.

## 1. Introduction

Cardiovascular disease (CVD) is a leading cause of death worldwide. The World Health Organization [1] reported that, in 2015, an estimated 17.7 million people died of CVD, accounting for 31% of deaths globally. For the prevention of CVD, reducing the effects of modifiable physiological risk factors for CVD, such as high blood pressure, diabetes, overweightness or obesity, and high blood cholesterol, through lifestyle and behavioral changes is crucial [1,2]. Physical inactivity, smoking, an unhealthy diet, and sedentary behaviors have been considered to be associated with increased risk of CVD and its related physiological risk factors [3]. WHO and a number of studies have indicated that workplace health promotion programs can be beneficial to employees’ health [4,5]. In addition, long working hours, shift work, and psychosocial stress at work are the risk factors of CVD [6,7]. Therefore, government regulators and employers need to take action to prevent CVD among workers.

Sedentary behavior is any waking behavior characterized by an energy expenditure of ≤1.5 metabolic equivalents while in a sitting, reclining, or lying posture [8]. Some studies have provided evidence of a relationship between sedentary behavior and sociodemographic characteristics [9,10]. Sedentary behavior was reported to be independently associated with CVD risk, regardless of physical activity levels [11]. Sedentary behavior may occur in a variety of contexts, including during home-based, leisure, occupational, or transport activities [12]. Many studies have provided evidence of a relationship between sedentary behavior and CVD risk [13,14,15]. Although a review indicated that a greater amount of time spent using technology or sitting is associated with an increased risk of CVD [16], the relationships of other domain-specific sedentary behaviors derived from different contexts with CVD risk factors remain unclear. Some researchers have suggested that different domains of sedentary behavior (i.e., occupational, transport, and leisure) may have distinct associations with health outcomes [17,18]. To design effective and tailored intervention strategies for preventing CVD, identifying the roles of different elements of sedentary behavior in CVD risk factors is crucial.

Innovative and high-tech industries have become increasingly successful in modern society [19], and employees working in high-tech companies are more likely to have reduced demands for physical activity and increased demands for prolonged and uninterrupted sedentary behavior (i.e., programming and administrative tasks) with respect to their occupational activity [20]. These employees may also be required to spend more time exhibiting sedentary behavior in other domains (e.g., leisure and transport) in addition to workplace sitting time as a result of high levels of exposure to high-tech products. In this context, Taiwan offers a unique research opportunity because the increasing global demand for mobile devices and access to the Internet has promoted the growth of Taiwan’s high-tech industries [21]. To improve the prevention of CVD for this at-risk population of high-tech company employees, this study examined the overall and domain-specific (i.e., occupational, transport, and leisure-time) sedentary behavior of individuals in a sample from Taiwan and the associations of these behaviors with CVD risk factors. We hypothesized that high-tech company employees who engaged in higher overall and domain-specific sedentary behavior were more likely to have CVD risk factors.

## 2. Materials and Methods

### 2.1. Participants

Participants were recruited from June to September 2018 from three Science Parks in Taiwan: Neihu Technology Park, Nangang Software Park, and Oriental Science Park. There were approximately 200,000 people working at these three Science Parks. We selected six companies (about 2700 employees as the target population). Only people aged between 18 to 65 years who were full-time or part-time employed at high-tech companies were recruited. We invited about 1600 employees to join this study via email. Finally, a total of 368 participants responded to our invitation, and 363 of these completed the questionnaire. Convenience sampling was utilized. This study was approved by the National Taiwan Normal University Human Research Ethics Committee (REC 201803HM020).

### 2.2. Sedentary Behavior

The survey components were based on the Sedentary Behavior Questionnaire for the Elderly of Taiwan. The questionnaire had adequate test–retest (r = 0.74) and concurrent validity (r = 0.52, *p* < 0.001) levels [22]. This study evaluated occupational (working or volunteering), transport or driving, leisure-time, and overall sedentary behaviors (e.g., watching TV, videos, or DVDs, using the computer or Internet, reading, engaging in social communication, and eating). Participants self-reported these sedentary behaviors in response to the following questions: “How many days did you engage in the behavior in the last 7 days?” and “How many hours and minutes did you spend engaging in this sedentary behavior on such a day?”. The sum of the time spent on engagement in occupational, transport, leisure-time, and overall sedentary behaviors was presented as minutes per day. According to previous studies [23,24,25], we used the cut-off value of 2 h/day to dichotomize leisure-time computer use and TV viewing time into “high (≥2 h/day)” and “low (<2 h/day)” groups. Other sedentary behaviors were categorized as “high” and “low” groups using median minutes per day.

### 2.3. CVD Risk Factors

Traditional CVD risk factors include waist circumference, body mass index (BMI), systolic and diastolic blood pressure, total and high-density lipoprotein cholesterol levels, and glycated hemoglobin levels [26,27]. CVD risk factors in the present study included (1) BMI (body weight and height were converted into BMI values); (2) blood pressure (normal range was 90/60–140/90 mmHg); (3) waist circumference (normal range was <80 cm for women and <90 cm for men); (4) total cholesterol level (normal range was <200 mg/dL); (5) triglyceride level (normal range was <150 mg/dL, 150–200 mg/dL was borderline high); and (6) fasting blood sugar (normal range was 70–110 mg/dL). All surveyed employees received mandatory physical examinations at their workplace, thus, we asked participants to report the information of waist circumference, body mass index, systolic and diastolic blood pressure, total and high-density lipoprotein cholesterol levels, and fasting blood sugar levels on the basis of most recent physical examination results.

### 2.4. Sociodemographic Factors

In the survey, participants reported their age, sex, marital status (married/divorced/widowed/never married), education level, number of children in the household, nature of their occupation and their employment position, and the presence of any chronic disease (diagnosed by a doctor). Education level was categorized as less than university level (high school/vocational school), university, and master’s level and beyond. Presence of children in the household was categorized as yes, no, or expecting. Nature of work was categorized as staff or supervisor, and work position was categorized as administrative work, research and development (R&D) engineering, on-site operation, and other.

### 2.5. Statistical Analysis

The data of 363 employees who provided information for the study were analyzed. A binary logistic regression was used to estimate odds ratios (ORs) and 95% confidence intervals (CIs) for the associations between total and domain-specific (occupational, transport, leisure-time) sedentary activity and CVD risk factors. Three models were analyzed. Model 1 was the crude model. Model 2 was adjusted for gender and age. Model 3 was further adjusted for gender, age, marital status, educational level, number of children, work type, and work position. All analyses were performed with SPSS Version 23, and the level of significance was set at *p* < 0.05.

## 3. Results

Table 1 presents the sociodemographic characteristics in the total sample. Overall, the mean age (standard deviation) of the respondents was 37.4 (±7.2) years. In total, 56.7% of the respondents were men, 62.5% were younger than 39 years, 54.3% were married, 55.9% had an educational level less than university, 55.9% had no children, 59.8% worked in R&D, and 63.9% were staff. There were four participants (1.1%) who had a history of CVD.

Table 2 lists the means and percentages of the sedentary behavior variables. Of the total sitting time, 38%, 30%, 15%, 9%, and 7% comprised occupational, other, leisure-time computer use, transport, and TV viewing, respectively. The mean (standard deviation) overall sedentary time was 6088.16 (2687.77) min/week. Of the five domains, the average amounts of time spent at work (2335.54 ± 777.18 min/week), on other activities (1849.74 ± 1567.2 min/week), using a computer during leisure time (925.21 ± 963.69 min/week), on transport (570.87 ± 688.72 min/week), and on watching TV (406.60 ± 437.58 min/week) are shown in Table 3.

Table 3 also lists the mean of sedentary behavior patterns without and with CVD risk. The mean (standard deviation) of overall sedentary time without CVD risk and with CVD risk were 6368.68 (3394.912) and 5940.82 (2222.79) min/week, respectively. Of the five domains, the average amounts of time spent at work (occupational sitting) without and with CVD risk, respectively, were 2323.72 (749.72) and 2341.74 (792.70) min/week; on watching TV were 370.16 (393.69) and 426.04 (458.57) min/week; on other activities (other sitting) were 60.24 (118.63) and 70.81 (195.52) min/week; leisure time using a computer were 1001.66 (1349.89) and 885.06 (678.54) min/week; on transport were 600.67 (799.21) and 555.21 (624.18) min/week. Significant differences between CVD risk groups were only observed in leisure time using a computer (*p* < 0.05).

Table 4 presents the ORs for the associations between overall and domain-specific (i.e., occupational, transport, and leisure-time) sedentary behavior and CVD risk factors among high-tech company employees. In model 1 and model 2, no significant associations between sedentary behaviors and CVD risk factor were observed. In model 3 (full-adjusted model), respondents who reported greater amounts of time spent using computers were more likely to have more than one CVD risk factor (OR = 1.80, 95% CI = 1.08–3.00; *p* = 0.02) compared with those with lower amounts of computer use time. No significant association with CVD risk factors was observed for total sedentary time (*p* = 0.10), occupational sitting time (*p* = 0.43), TV viewing time (*p* = 0.14), or transport-related sitting time (*p* = 0.75).

## 4. Discussion

To the best of our knowledge, this is the first study to examine the associations of overall and domain-specific (i.e., occupational, transport, and leisure-time) sedentary behavior with CVD risk factors in the specific work population of high-tech company employees. The main finding of this study is that spending more than 2 h per day on leisure-time computer use was associated with a greater likelihood of having more than one CVD risk factor among high-tech company employees, after physical activity and other confounders were controlled for. No significant associations were observed for total sedentary time, occupational sitting, transport-related sitting, and TV viewing. Therefore, these findings may have profound implications for workplace intervention designers and policymakers; specifically, effective strategies should prioritize the reduction of leisure-time computer use rather than that of occupational or transport sitting to reduce the risk of CVD among high-tech company employees. The government or employers can promote workers to attend more activities after work, such as exercise programs or social activity, in order to replace their time spent on leisure-time computer use.

Consistent with several other studies [17,28,29], leisure-time sedentary behavior poses greater risk to cardio-metabolic health than do total sedentary time and other domains of sedentary behavior. Our results also reveal that between the two types of leisure-time sedentary behavior, only leisure-time computer use, and not TV viewing, was associated with an increased likelihood of having CVD risk factors. Although TV viewing is considered a dominant leisure-time sedentary behavior with negative effects on cardio-metabolic health [22,30,31], it is possible that with the evolution of modern technology, computer use (including Internet and smartphone use) has become a more prevalent leisure-time activity in the adult population, which may increase the risk of developing CVD risk factors. Indeed, the participants of this study spent higher time on leisure-time computer use (925.21 ± 963.69 min/week) than TV viewing (406.60 ± 437.58 min/week). Thus, possible explanations for the positive associations between leisure-time computer use and CVD risk factors are that leisure-time computer use is a prolonged sedentary activity characterized by uninterrupted breaks and low energy expenditure [32,33] and that it is possibly related to other unhealthy behaviors, such as unhealthy eating and watching TV [34]. Thus, our findings may highlight the value of limiting leisure-time computer use to less than 2 h per day among high-tech company employees and may provide evidence for future guidelines on CVD prevention.

Another critical finding of this study is that no significant associations were observed between CVD risk factors and total, occupational, or transport-related sitting among high-tech company employees. There are several possible explanations for this result. First, regarding total sedentary time, other studies have reported that specific domains of sedentary behavior have different associations with health outcomes [17,18,35]. Therefore, the effects of domain-specific sedentary behavior may be negligible in total sedentary time with respect to CVD risk factors. Second, regarding transport-related sitting, compared with Western countries, the amount of time spent in transport may occupy a smaller portion of daily sedentary time for the participants in our study, thus explaining its lack of association with CVD risk factors. Third, in contrast to several related studies that have reported a positive relationship between occupational sitting and CVD risk factors [36,37,38], our results show no significant associations. This inconsistency could be attributable to our cross-sectional design and the lack of consideration for breaks in occupational sedentary behavior. Therefore, future prospective studies using both objective and subjective measures should further explore this topic to confirm our results.

Several limitations of the present study should be considered. First, because of the cross-sectional design of this study, a causal link between sedentary behavior and CVD risk cannot be assumed. Second, several confounding factors, such as smoking, dietary behavior, and alcohol consumption, were not included in the present study. Third, in our study, we examined the CVD risk factors all together but not one by one due to the limited number of each risk factor. Fourth, the measurements in the present study, including sedentary behavior and status of CVD risk factors, were self-reported and could be subject to bias. In addition, we measured sedentary behavior only in the previous seven days, thus limiting the ability to assess long-term lifestyles. Finally, since gender is an important moderator between sedentary behavior and CVD risk factors, this study is limited in not examining these associations by gender. Future studies are warranted to examine gender-specific associations between sedentary behavior and CVD risk factors among high-tech company employees.

## 5. Conclusions

Despite the cross-sectional nature and small sample size in this study, our results support the view that workers in high-tech companies need to be aware that sedentary behaviors, particularly leisure-time computer use, could be one of the risk factors of CVD, in the context of Taiwan. These findings may have considerable implications for local policymakers and designers of workplace interventions. In addition to programs of exercise and healthy dieting, priority should be given to developing effective strategies for limiting leisure-time computer use.

## Figures and Tables

**Table 1 ijerph-17-03353-t001:** Demographic characteristics of high-tech company employees.

Variable	Category	Total Sample		Without CVD Risk Factors		With CVD Risk Factors	
		*n* = 363	%	*n* = 125	%	*n* = 238	%
Gender	Female	157	43.3	65	52.0%	92	38.7%
	Male	206	56.7	60	48.0%	146	61.3%
Age level	Mean 37.7y (SD 7.1)						
	<40 year	227	62.5	89	71.2%	138	58.0%
	≧40 year	136	37.5	36	28.8%	100	42.0%
Marital status	Unmarried	166	45.7	65	52.0%	101	42.4%
	Married	197	54.3	60	48.0%	137	57.6%
Number of children	No child	203	55.9	74	59.2%	129	54.2%
	have child	160	44.1	51	40.8%	109	45.8%
Job grade	Staff	232	63.9	83	66.4%	149	62.6%
	Supervisor	131	36.1	42	33.6%	89	37.4%
Position	other work	146	40.2	53	42.4%	93	39.1%
	R&D	217	59.8	72	57.6%	145	60.9%
Education level	University	203	55.8	74	59.2%	129	54.2%
	Master degree or Dr.	160	44.1	51	40.8%	109	45.8%

CVD, cardiovascular disease; SD, standard deviation.

**Table 2 ijerph-17-03353-t002:** Descriptive statistical summary of time spent on various sedentary behaviors.

Variable	CVD	Average Time (SD)
Occupational sitting	no	2346.35 (768.73)
	yes	1762.50 (1239.20)
TV viewing	no	404.53 (437.30)
	yes	480.00 (379.47)
Leisure-time computer use	no	920.53 (955.96)
	yes	1560.00 (1409.68)
Transport-related sitting	no	565.62 (682.43)
	yes	1012.50 (819.89)
Other sitting time	no	1840.65 (1545.56)
	yes	2855.00 (3069.64)
Total sitting time	no	6077.67 (2652.96)
	yes	7670.00 (4825.87)

SD, standard deviation; min/week.

**Table 3 ijerph-17-03353-t003:** The mean of sedentary behavior patterns by with and without CVD risk.

Variables	Total Sample	Without CVD Factor	With CVD Factors			
	*n* = 363 (100%) Average Time (SD)	*n* = 125 (34.4%) Average Time (SD)	*n* = 238 (65.6%) Average Time (SD)	*t*	*df*	*P*
Occupational sitting	2335.54 (777.18)	2323.72 (749.72)	2341.74 (792.70)	−0.210	361	0.332
TV viewing	406.8 (437.58)	370.16 (393.69)	426.04 (458.57)	−1.157	361	0.089
Leisure-time computer use	925.21 (963.69)	1001.66 (1349.89)	885.06 (678.54)	1.096	361	0.002
Transport-related sitting	570.87 (688.72)	600.67 (799.21)	555.21 (624.18)	0.597	361	0.436
Other sitting	1849.74 (1567.2)	60.24 (118.63)	70.81 (195.52)	0.549	−0.553	361
Total sitting	6088.16 (2687.77)	6368.68 (3394.912)	5940.82 (2222.79)	0.017	1.443	361

SD, standard deviation; min/week.

**Table 4 ijerph-17-03353-t004:** Summary of the logistic regression analysis of time spent engaged in sedentary behavior and cardiovascular disease risk factors.

Variable	Category	Model 1			Model 2			Model 3		
		CVD Risk Factors			CVD Risk Factors			CVD Risk Factors		
		OR	95%CI	*p*-Value	OR	95%CI	*p*-Value	OR	95%CI	*p*-Value
**Total sitting time**	Low	1.00 (ref.)			1.00 (ref.)			1.00 (ref.)		
	High	0.84	0.92–2.45	0.53	0.95	0.53–1.67	0.85	1.50	0.92–2.45	0.10
**Occupational sitting**	Low	1.00 (ref.)			1.00 (ref.)			1.00 (ref.)		
	High	0.82	0.51–1.32	0.42	0.82	0.50–1.33	0.42	0.81	0.86–2.89	0.43
**TV viewing**	Low	1.00 (ref.)			1.00 (ref.)			1.00 (ref.)		
	High	1.61	0.89–2.93	0.12	1.47	0.80–2.71	0.22	1.57	0.86–2.89	0.14
**Leisure-time computer use**	Low	1.00 (ref.)			1.00 (ref.)			1.00 (ref.)		
	High	1.48	0.90–2.42	0.12	1.62	0.97–2.71	0.07	1.80	1.08–3.00	0.02 *
**Transport-related sitting**	Low	1.00 (ref.)			1.00 (ref.)			1.00 (ref.)		
	High	1.36	0.86–2.17	0.19	1.30	0.81–2.08	0.28	1.08	0.65–1.78	0.75

OR, odds ratio; CI, confidence interval; Notes: * *p*-value < 0.05; Model 1 is the crude model; Model 2 adjusted for gender and age; Model 3 adjusted for gender, age, marital status, educational level, number of children, work type, and work position.

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
