# Peer review of "Occupational, Transport, Leisure-Time, and Overall Sedentary Behaviors and Their Associations with the Risk of Cardiovascular Disease among High-Tech Company Employees"

_ijerph, 2020, doi:10.3390/ijerph17103353_

Round 1
Reviewer 1 Report
Liu et al. examined the overall and domain-specific (i.e., occupational, transport, and leisure-time) sedentary behavior of individuals in a sample from Taiwan and the associations of these 81 behaviors with CVD risk factors. The main finding of this study is that spending more than 2 hours per day on leisure-time computer use was associated with a greater likelihood of having more than one CVD risk factor among high-tech company employees, after physical activity and other confounders were controlled for. No significant associations were observed for total sedentary time, occupational sitting, transport-related sitting, and TV viewing. These findings in this study is interesting. However, the following comments and addressing them will strengthen the manuscript.
The authors should address whether the participants had CVD before the survey.
2. The lifestyle of the participants such as smoking and drinking habits should be addressed.
3. The impact of gender on the association of CVD and overall/sedentary behaviors should also addressed.
4. The authors presented overall/sedentary time/week. It is not clear how long these behaviors lasted (e.g., months or years?).
Author Response
Occupational, transport, leisure-time, and overall sedentary behaviors and their associations with the risk of cardiovascular disease among high-tech company employees.
Manuscript ID: ijerph-673607
Dear Reviewer 1,
The authors wish to thank the reviewer for reading our manuscript so thoroughly and providing such constructive feedback. The quality of the manuscript has certainly improved as a result of these comments. Our responses and the necessary changes are included here and in the revised manuscript. We have listed the comments followed by our responses below. The revised sentences are highlighted in yellow.
General Comments
Liu et al. examined the overall and domain-specific (i.e., occupational, transport, and leisure-time) sedentary behavior of individuals in a sample from Taiwan and the associations of these 81 behaviors with CVD risk factors. The main finding of this study is that spending more than 2 hours per day on leisure-time computer use was associated with a greater likelihood of having more than one CVD risk factor among high-tech company employees, after physical activity and other confounders were controlled for. No significant associations were observed for total sedentary time, occupational sitting, transport-related sitting, and TV viewing. These findings in this study is interesting. However, the following comments and addressing them will strengthen the manuscript.
Response: Thanks for your comments.
Query:
- The authors should address whether the participants had CVD before the survey.
Response 1:
Thank you very much for your valuable comments. We have added this information in the Results accordingly. (page 8, line 178)
four participants (1.1%) who had the history of CVD.
- The lifestyle of the participants such as smoking and drinking habits should be addressed.
Response 2:
Thank you very much for your comments. We have added this as a limitation and revised accordingly. (page 11, line 268-269)
Second, several confounding factors, such as smoking, dietary behavior, and alcohol consumption, were not included in the present study.
- The impact of gender on the association of CVD and overall/sedentary behaviors should also addressed.
Response 3:
Thank you very much for your comments. We agree with your comments. The key reason that we did not divide our sample by gender is because the main purpose of this study is to provide preliminary findings on the association between overall/domain-specific sedentary behavior and CVD risk factors in overall sample. We have added this important issue for future studies. (page 11, line 275-278)
Finally, since gender is an important moderator between sedentary behavior and CVD risk factors, this study is limited examining these association by gender. Future studies are warranted to examine gender-specific association between sedentary behavior and CVD risk factors among high-tech company employees.
- The authors presented overall/sedentary time/week. It is not clear how long these behaviors lasted (e.g., months or years?).
Response 4:
Thank you very much for your comments. We have added this as a limitation and revised accordingly. (page 11, line 273-275)
In addition, we measured sedentary behavior only in past seven day, which is limited to assess their long-term lifestyle.
Thank you again for your time and consideration. We hope that you find these adjustments satisfactory and that the revised version will be acceptable for publication in Special issue.
Sincerely yours,
The Authors
Reviewer 2 Report
Liu ML et al. conducted a questionnaire survey of high-tech company employees to assess if their overall sedentary behaviors can increase CVD risk factors among them. The results indicated greater amounts of computer use time during leisure-time likely to have more than one CVD risk factor compared with those with lower amounts of computer use time. The author gave a conclusion that limiting leisure-time computer use rather than occupational or transport sitting can be an effective strategy to reduce the risk of CVD among high-tech company employees in Taiwan. The concept could be helpful to develop considerable guideline for the associated workers. However, there are several concerns that needed to make changes.
Demographic characteristics of high-tech company employees in Table 1 should be also show in the comparison with or without CVD risk factors. In Table 2, each sedentary behavior should be checked in univariate analyses as an independent variable of having more than one CVD risk factor. The results based on self-reported data never reflect actual occurrence of cardiovascular events. The point is, just showing that not making exercise a habit can increase the number of risk factors for lifestyle related disease. It’s too much to say that limiting leisure-time computer use rather than occupational or transport sitting can reduce the risk of CVD among high-tech company employees. The conclusion of this study should be corrected to follow the exact results of increasing the related risk factors. The author referred in Results that more than 2 hours per day during their leisure-time likely to have CVD risk factors. First, the cutoff value should be argued in receiver operating characteristic with explanatory covariates. Respective CVD risk factors in the present study can be further examined in two groups divided with the cutoff value of computer use during leisure-time.
Author Response
Occupational, transport, leisure-time, and overall sedentary behaviors and their associations with the risk of cardiovascular disease among high-tech company employees.
Manuscript ID: ijerph-673607
Dear Reviewer 2,
The authors wish to thank the reviewer for reading our manuscript so thoroughly and providing such constructive feedback. The quality of the manuscript has certainly improved as a result of these comments. Our responses and the necessary changes are included here and in the revised manuscript. We have listed the comments followed by our responses below. The revised sentences are highlighted in yellow.
General Comments
Liu ML et al. conducted a questionnaire survey of high-tech company employees to assess if their overall sedentary behaviors can increase CVD risk factors among them. The results indicated greater amounts of computer use time during leisure-time likely to have more than one CVD risk factor compared with those with lower amounts of computer use time. The author gave a conclusion that limiting leisure-time computer use rather than occupational or transport sitting can be an effective strategy to reduce the risk of CVD among high-tech company employees in Taiwan. The concept could be helpful to develop considerable guideline for the associated workers. However, there are several concerns that needed to make changes.
Response: Thank you very much for your positive comments.
Query
- Demographic characteristics of high-tech company employees in Table 1 should be also show in the comparison with or without CVD risk factors.
Response 1:
Thank you very much for your valuable comments. We had add this information in Table 1. (page 16)
|
Variable |
Category |
Total sample |
Without CVD risk factors |
With CVD risk factors |
|||
|
n=363 |
% |
n=125 |
% |
n=238 |
% |
||
|
Gender |
Female |
157 |
43.3 |
65 |
52.0% |
92 |
38.7% |
|
Male |
206 |
56.7 |
60 |
48.0% |
146 |
61.3% |
|
|
Age level |
MD 37.7y (SD 7.1) |
|
|
|
|
|
|
|
<40 year |
227 |
62.5 |
89 |
71.2% |
138 |
58.0% |
|
|
≧ 40 year |
136 |
37.5 |
36 |
28.8% |
100 |
42.0% |
|
|
Marital status |
Unmarried |
166 |
45.7 |
65 |
52.0% |
101 |
42.4% |
|
Married |
197 |
54.3 |
60 |
48.0% |
137 |
57.6% |
|
|
Number of child |
No child |
203 |
55.9 |
74 |
59.2% |
129 |
54.2% |
|
have child |
160 |
44.1 |
51 |
40.8% |
109 |
45.8% |
|
|
Job grade |
Staff |
232 |
63.9 |
83 |
66.4% |
149 |
62.6% |
|
Supervisor |
131 |
36.1 |
42 |
33.6% |
89 |
37.4% |
|
|
Position |
other work |
146 |
40.2 |
53 |
42.4% |
93 |
39.1% |
|
R&D |
217 |
59.8 |
72 |
57.6% |
145 |
60.9% |
|
|
Education level |
University |
203 |
55.8 |
74 |
59.2% |
129 |
54.2% |
|
Master degree or Dr. |
160 |
44.1 |
51 |
40.8% |
109 |
45.8% |
|
- In Table 2, each sedentary behavior should be checked in univariate analyses as an independent variable of having more than one CVD risk factor.
Response 2:
Thank you very much for your valuable comments. We had add this information in Table 3. (page 18)
Table 3. Means of sedentary behavior patterns for individuals without CVD risk factors and those with one CVD risk factor or more.
|
Variables |
Total sample |
Without CVD factor |
With CVD factors |
|
|
|
|
n=363 (100) |
n=125 (34.4) |
n=238 (65.6) |
t |
df |
p |
|
|
Occupational sitting |
2335.54 (777.18) |
2323.72 (749.72) |
2341.74 (792.70) |
-.210 |
361 |
.332 |
|
TV viewing |
406.8 (437.58) |
370.16 (393.69) |
426.04 (458.57) |
-1.157 |
361 |
.089 |
|
Leisure_time computer use |
925.21 (963.69) |
1001.66 (1349.89) |
885.06 (678.54) |
1.096 |
361 |
.002 |
|
Transport-related sitting |
570.87 (688.72) |
600.67 (799.21) |
555.21 (624.18) |
.597 |
361 |
.436 |
|
Other sitting |
1849.74 (1567.2) |
60.24 (118.63) |
70.81 (195.52) |
.549 |
-.553 |
361 |
|
Total sitting |
6088.16 (2687.77) |
6368.68 (3394.912) |
5940.82 (2222.79) |
.017 |
1.443 |
361 |
- The results based on self-reported data never reflect actual occurrence of cardiovascular events. The point is, just showing that not making exercise a habit can increase the number of risk factors for lifestyle related disease. It’s too much to say that limiting leisure-time computer use rather than occupational or transport sitting can reduce the risk of CVD among high-tech company employees. The conclusion of this study should be corrected to follow the exact results of increasing the related risk factors.
Response 3:
Thank you very much for your valuable comments. We corrected the information of conclusion section. (page 12, line 281-288)
Our results support that workers in high-tech companies need to aware sedentary behaviors could be one of the risk factors of CVD, especially leisure-time computer use. These findings may have considerable implications for policymakers and designers of workplace interventions that priority should be given to developing effective strategies for limiting leisure-time computer use. For the government and employers, it is critical to promote and support the health of workers Many workplaces have several activities to prevent CVD, such as programs of exercise, body weight control and healthy diet. Our findings may suggest these health-promoting programs should also provide strategies to limit leisure-time computer use for the prevention of CVD.
- The author referred in Results that more than 2 hours per day during their leisure-time likely to have CVD risk factors. First, the cutoff value should be argued in receiver operating characteristic with explanatory covariates. Respective CVD risk factors in the present study can be further examined in two groups divided with the cutoff value of computer use during leisure-time.
Response 4:
Thank you very much for your valuable comments. First of all, we have added several references to support the cut-off value of leisure-time computer use and TV viewing (2 hour/day). And other sedentary behaviors were divided by the median value. We have added this in the Method section (page 6, line 132-137)
After consulting previous studies (Grøntved & Hu, 2011; Jakes et al., 2003; Thomee et al., 2015), we decided on a threshold of 2 h/day to define two groups: “highly sedentary” (≥2 h/day) and “slightly sedentary” (<2 h/day). Other sedentary behaviors were categorized as being part of the “highly sedentary” or “slightly sedentary” group in terms of the median minutes per day spent engaged in that behavior.
Second, we acknowledge that we put all CVD risk factors together for analyses as a major limitation in this study and add our reason in the limitation.
In our study, we examined the CVD risk factors all together but not one by one due to the limited number of each risk factor.
Thank you again for your time and consideration. We hope that you find these adjustments satisfactory and that the revised version will be acceptable for publication in Special issue.
Sincerely yours,
The Author
Reviewer 3 Report
Occupational, Transport, leisure-time and Overall Sedentary Behaviors and Their Associations with the Risk of Cardiovascular Disease among High5 Tech Company eEmployees.
Thank you for the opportunity to review the article. The article examines an important topic, I mean, the associations of overall and domain specific (i.e., occupational, transport, and leisure-time) sedentary behaviors with CVD risk factors among high-tech company employees in Taiwan. I think the main limitations of the article are: the total sample, the cross-sectional analyses and self-reported DCV risk factors (height and weight, blood pressure, blood sugar, and total cholesterol levels) and were not included important confounders in the analyses
Abstract: please, remove the expression “were more likely to have”. Unfortunately, in cross-sectional analyses we have limitation on risk estimation and we can observe association.
The introduction section is well written and justified
Methods: was the 368 the total population of workers from three Science Parks in 87 Taiwan? This is not clear to me. The authors wrote “Convenience sampling was utilized”, but there is not details about the recruitment.
The sentences: “Many studies have provided evidence of a relationship between sedentary behavior 109 and CVD risk (Cabanas-Sánchez, Guallar-Castillón, Higueras-Fresnillo, Rodríguez110 Artalejo, & Martínez-Gómez, 2018; Hallman et al., 2019; Young et al., 2016)” and “Some studies have provided evidence of a relationship between sedentary behavior and sociodemographic characteristics (Celis-Morales C et al., 2015; Lakerveld J et al., 127 2017)” should be moved to the introduction or discussion section.
Please, provide information about how the data of waist circumference, body mass index, systolic and diastolic blood pressure, total and high-density lipoprotein cholesterol levels, and glycated hemoglobin levels were collected.
Why all risk factors were put together in the analyses? Maybe could be interesting analyses the association between sedentary behaviors and each risk factor in separate.
Please inform how the categories (high and low) of each sedentary behavior was formulated? Based on mean, median…?
Results: page 6, line 150 review the 63.9.0% information
Could the authors include leisure physical activity as a covariates in the analyses? I think that variable could be one of the main confounders of analyses.
Author Response
Occupational, transport, leisure-time, and overall sedentary behaviors and their associations with the risk of cardiovascular disease among high-tech company employees.
Manuscript ID: ijerph-673607
Dear Reviewer 3,
The authors wish to thank the reviewer for reading our manuscript so thoroughly and providing such constructive feedback. The quality of the manuscript has certainly improved as a result of these comments. Our responses and the necessary changes are included here and in the revised manuscript. We have listed the comments followed by our responses below. The revised sentences are highlighted in yellow.
General Comments
Thank you for the opportunity to review the article. The article examines an important topic, I mean, the associations of overall and domain specific (i.e., occupational, transport, and leisure-time) sedentary behaviors with CVD risk factors among high-tech company employees in Taiwan. I think the main limitations of the article are: the total sample, the cross-sectional analyses and self-reported DCV risk factors (height and weight, blood pressure, blood sugar, and total cholesterol levels) and were not included important confounders in the analyses.
Response: Thanks for your comments.
Query
- Abstract: please, remove the expression “were more likely to have”. Unfortunately, in cross-sectional analyses we have limitation on risk estimation and we can observe association.
Response 1:
Thank you very much for your valuable comments.
We have removed this information in the Results accordingly. (page 2, line 42-43)
- The introduction section is well written and justified.
Response 2:
Thanks for your positive comments.
- Methods: was the 368 the total population of workers from three Science Parks in 87 Taiwan? This is not clear to me. The authors wrote “Convenience sampling was utilized”, but there is not details about the recruitment.
Response 3:
Thanks for your comments. We add the detailed information in the Method section. (page 5, line 113-117)
We selected six high-tech companies (2,700 employees in total as the target population), and only people between the ages of 18 and 65 years who were full-time or part-time employees at these companies were recruited. We invited approximately 1,600 employees by email to join this study, among whom 368 responded to our invitation and 363 completed the questionnaire.
- The sentences: “Many studies have provided evidence of a relationship between sedentary behavior 109 and CVD risk (Cabanas-Sánchez, Guallar-Castillón, Higueras-Fresnillo, Rodríguez110 Artalejo, & Martínez-Gómez, 2018; Hallman et al., 2019; Young et al., 2016)” and “Some studies have provided evidence of a relationship between sedentary behavior and sociodemographic characteristics (Celis-Morales C et al., 2015; Lakerveld J et al., 127 2017)” should be moved to the introduction or discussion section.
Response 4:
Thanks for your comments.
We have moved this information in the Background accordingly. (page 3)
- Please, provide information about how the data of waist circumference, body mass index, systolic and diastolic blood pressure, total and high-density lipoprotein cholesterol levels, and glycated hemoglobin levels were collected.
Response 5:
Thanks for your comments. We add this important information into Method section. (page 6, line 149-153)
Because all employees in this study had to undergo a mandatory workplace physical examination, data on waist circumference, BMI, systolic and diastolic blood pressure, total and high-density lipoprotein cholesterol levels, and fasting blood sugar levels were based on the most recent results of a participant’s physical examination, as reported by the participant.
- Why all risk factors were put together in the analyses? Maybe could be interesting analyses the association between sedentary behaviors and each risk factor in separate.
Response 6:
Thank you very much for your valuable comments. And we add this as a limitation of this study. (page 11, line 269-271)
Third, we examined CVD risk factors together and not individually because of the limited number of participants with any one risk factor.
- Please inform how the categories (high and low) of each sedentary behavior was formulated? Based on mean, median…?
Response 7:
Thanks for your comments. We have added this important information accordingly. (page 6, line 132-137)
After consulting previous studies (Grøntved & Hu, 2011; Jakes et al., 2003; Thomee et al., 2015), we decided on a threshold of 2 h/day to define two groups: “highly sedentary” (≥2 h/day) and “slightly sedentary” (<2 h/day). Other sedentary behaviors were categorized as being part of the “highly sedentary” or “slightly sedentary” group in terms of the median minutes per day spent engaged in that behavior.
- Results: page 6, line 150 review the 63.9.0% information.
Could the authors include leisure physical activity as a covariates in the analyses? I think that variable could be one of the main confounders of analyses.
Response 8:
Thanks for your comments. First of all, we have corrected this wrong information. (page 8, line 181)
Second, we did not measure physical activity in this study and it could be a key confounder of this study. We have added this as a major limitation of this study. (page 11, line 268-269)
Second, several confounding factors, such as smoking habit, dietary behavior, and alcohol consumption, were not included in the present study.
Thank you again for your time and consideration. We hope that you find these adjustments satisfactory and that the revised version will be acceptable for publication in Special issue.
Sincerely yours,
The Authors
Reviewer 4 Report
Thank you very much for the opportunity to review your work. I think it is a topic of interest and that it must be greatly improved to be published in the magazine. They should improve some fundamental labor issues:
1. Review the theoretical foundation is quite weak. They should at least name some aspects such as burnout and its relation to workers' health.
2. Improving the previous aspect, they will be able to formulate the necessary hypotheses in the investigation and that currently do not appear, together with the general and specific objective.
3. Specify the selection criteria for the sample.
4. References are not in the journal format.
5. Indicate the reliability and validity of the instruments used in the sample.
6. They should work on deeper statistical analyzes with more complex regression analyzes or mediation models, for example, trying to explain the variance of the research objective variable. In the regression analysis that is performed it is not shown in a table of variables entering the model, nor the results obtained.
7. The discussion should include practical applications of the study, as well as future lines of research.
8. The conclusion is limited to 4 lines, which reflects the poor deepening of the contents made by the authors.
Author Response
Occupational, transport, leisure-time, and overall sedentary behaviors and their associations with the risk of cardiovascular disease among high-tech company employees.
Manuscript ID: ijerph-673607
Dear Reviewer 4,
The authors wish to thank the reviewer for reading our manuscript so thoroughly and providing such constructive feedback. The quality of the manuscript has certainly improved as a result of these comments. Our responses and the necessary changes are included here and in the revised manuscript. We have listed the comments followed by our responses below. The revised sentences are highlighted in yellow.
General Comments
Thank you very much for the opportunity to review your work. I think it is a topic of interest and that it must be greatly improved to be published in the magazine. They should improve some fundamental labor issues:
Response: Thanks for your comments.
Query
- Review the theoretical foundation is quite weak. They should at least name some aspects such as burnout and its relation to workers' health.
Response 1:
Thank you very much for your valuable comments. We have added this information in background section. (page 3, line 62-68)
A healthy lifestyle that involves the avoidance of prolonged sitting can improve cardiovascular health. The WHO and researchers have indicated the benefits of workplace health promotion programs to employees’ health (Coffeng et al., 2017; Martin et al., 2015). In addition, long working hours, irregular shifts, and psychosocial stress at work are risk factors of CVD (Fishta and Backé 2015; Kivimäki et al. 2015). Therefore, measures for preventing CVD among workers must involve both the government and employers.
- Improving the previous aspect, they will be able to formulate the necessary hypotheses in the investigation and that currently do not appear, together with the general and specific objective.
Response 2:
Thank you very much for your valuable comments. We have added our hypotheses in Background section. (page 5, line 105-107)
We hypothesized that high-tech company employees who engage in greater overall and domain-specific sedentary behaviors are prone to having higher risk of CVD.
- Specify the selection criteria for the sample.
Response 3:
Thank you very much for your valuable comments. We have modified this information. (page 5, line 114-116)
Only people between the ages of 18 and 65 years who were full-time or part-time employees at these companies were recruited.
- References are not in the journal format.
Response 4:
Thank you very much for your valuable comments. References format has been modified.
- Indicate the reliability and validity of the instruments used in the sample.
Response 5:
Thank you very much for your valuable comments. We add this information in Sedentary behavior section. (page 5, line 123-124)
The questionnaire had adequate test–retest (ρ = 0.74) and concurrent (ρ = 0.52, p < 0.001) validity (Ku, Sun, & Chen, 2016).
- They should work on deeper statistical analyzes with more complex regression analyzes or mediation models, for example, trying to explain the variance of the research objective variable. In the regression analysis that is performed it is not shown in a table of variables entering the model, nor the results obtained.
Response 6:
Thank you very much for your valuable comments. The reason we did not conduct deeper statistical analyzes is because the main purpose of this study is to provide preliminary findings on the association between overall/domain-specific sedentary behavior and CVD risk factors in overall sample. We have added this as a suggestion for future studies. (page 11, line 275-278)
Finally, although gender is an important moderator between sedentary behavior and CVD risk factors, we did not include it in our analysis. Future studies should investigate gender-specific associations between sedentary behavior and CVD risk factors among high-tech company employees.
- The discussion should include practical applications of the study, as well as future lines of research.
Response 7:
Thank you very much for your valuable comments. We had add this information in the discussion section. (page 9, line 221-224)
To ensure that employees spend less time on the computer during leisure time, employers or the government can encourage workers to participate in either physical activities outside of work, such as exercise programs, or social activities.
- The conclusion is limited to 4 lines, which reflects the poor deepening of the contents made by the authors.
Response 8:
Thank you very much for your valuable comments. We had add this information in the conclusion section. (page 12, line 281-288)
Our results indicate that workers in high-tech companies must be aware that sedentary behavior is a potential risk factor of CVD, especially that of leisure-time computer use. These findings have implications for policymakers and designers of workplace interventions, specifically that they should prioritize limiting leisure-time computer use among employees. For government and employers, the health of workers must be supported. Many workplaces organize activities to prevent CVD, such as exercise, weight control, and diet programs. Our findings suggest that these programs should also be aimed at limiting leisure-time computer use to prevent CVD.
Thank you again for your time and consideration. We hope that you find these adjustments satisfactory and that the revised version will be acceptable for publication in Special issue.
Sincerely yours,
The Authors
Round 2
Reviewer 1 Report
I have no further comments.
Author Response
Thank you for your review.
Reviewer 2 Report
The author responded to our concerns. The revised manuscript has reflected their responses and the necessary changes. The present study could provide clinical implications for the reduction of CVD risk factors among high-tech company employees although further well-designed studies need to corroborate the credibility.
Author Response
Thank you for your review.
Reviewer 4 Report
Authors must respond with changes in the work, as this is essential for acceptance.
They should work on deeper statistical analyzes with more complex regression analyzes or mediation models, for example, trying to explain the variance of the research target variable. In the regression analysis performed, it is not shown in a table of variables that enter the model, nor the results obtained.
They simply limit themselves to indicating that they will incorporate it as future lines.
Making these changes at work is critical to your acceptance.
Author Response
Dear Reviewer 4,
The authors wish to thank you for reading our manuscript so thoroughly and providing such constructive feedback. The quality of the manuscript has certainly improved as a result of these comments. Our responses and the necessary changes are included here and in the revised manuscript. We have listed the comments followed by our responses below. The revised sentences are highlighted in yellow.
General Comments
Authors must respond with changes in the work, as this is essential for acceptance.
Response: Thanks for your comments. We have listed our revisions as below:
Query
- They should work on deeper statistical analyzes with more complex regression analyzes or mediation models, for example, trying to explain the variance of the research target variable.
Response 1:
Thank you very much for your valuable comments. We have worked on deeper statistical analyzes using 3 logistic regression models. Model 1 is the crude model; Model 2 is adjusted for gender and age. Model 3 is full adjusted model (adjusted for gender, age, marital status, educational level, number of children, work type, and work position). The result showed that no significant association between sedentary behavior and CVD risk factors. The significant associations between leisure-time computer use and CVD risk factors were still observed only in full-adjusted model (model 3).
We have added this important information in the “Statistical analyses” (page 7, line 171-174), “Results” section (page 9, line 206-208) and Table 4 accordingly.
Three models were analyzed. Model 1 is the crude model. Model 2 adjusted for gender and age. Model 3 adjusted for gender, age, marital status, educational level, number of children, work type, and work position.
- In the regression analysis performed, it is not shown in a table of variables that enter the model, nor the results obtained.
Response 2:
Thank you very much for your valuable comments. We have listed this information in footnote of Table 4. (page 18-19)
Table 4. Summary of the logistic regression analysis of time spent engaged in sedentary behavior and cardiovascular disease risk factors
|
Variable |
category |
Model 1 |
|
Model 2 |
|
Model 3 |
|||||||||||
|
CVD risk factors |
CVD risk factors |
CVD risk factors |
|||||||||||||||
|
OR |
95%CI |
p-value |
OR |
95%CI |
p-value |
OR |
95%CI |
p-value |
|
||||||||
|
Total sitting time |
Low |
1.00 (ref.) |
1.00 (ref.) |
1.00 (ref.) |
|
||||||||||||
|
High |
0.84 |
0.92-2.45 |
0.53 |
0.95 |
0.53-1.67 |
0.85 |
1.50 |
0.92-2.45 |
0.10 |
|
|||||||
|
Occupational sitting |
Low |
1.00 (ref.) |
1.00 (ref.) |
1.00 (ref.) |
|
||||||||||||
|
High |
0.82 |
0.51-1.32 |
0.42 |
0.82 |
0.50-1.33 |
0.42 |
0.81 |
0.86-2.89 |
0.43 |
|
|||||||
|
TV viewing |
Low |
1.00 (ref.) |
1.00 (ref.) |
1.00 (ref.) |
|
||||||||||||
|
High |
1.61 |
0.89-2.93 |
0.12 |
1.47 |
0.80-2.71 |
0.22 |
1.57 |
0.86-2.89 |
0.14 |
|
|||||||
|
Leisure-time computer use |
Low |
1.00 (ref.) |
1.00 (ref.) |
1.00 (ref.) |
|
||||||||||||
|
High |
1.48 |
0.90-2.42 |
0.12 |
1.62 |
0.97-2.71 |
0.07 |
1.80 |
1.08-3.00 |
0.02* |
|
|||||||
|
Transport-related sitting |
Low |
1.00 (ref.) |
1.00 (ref.) |
1.00 (ref.) |
|
||||||||||||
|
High |
1.36 |
0.86-2.17 |
0.19 |
|
1.30 |
0.81-2.08 |
0.28 |
|
1.08 |
0.65-1.78 |
0.75 |
|
|||||
OR, odds ratio; CI, confidence interval
Notes: * p-value< 0.05, ** p-value< 0.01, *** p-value< 0.001.
Model 1 is the crude model.
Model 2 adjusted for gender and age.
Model 3 adjusted for gender, age, marital status, educational level, number of children, work type, and work position.
- They simply limit themselves to indicating that they will incorporate it as future lines.
Making these changes at work is critical to your acceptance.
Response 3:
Thank you again for your time and consideration. We hope that you find these adjustments satisfactory and that the revised version will be acceptable for publication.
